# CO$_2$-triggered reversible transformation of soft elastomers into rigid and highly fluorescent plastics

Yohei Miwa [1] ✉, Kazuma Okada [2], Takumi Hayashi[3], Kei Hashimoto [1], Hikaru Okubo[4], Hiroshi Takase[5], Katsuhiro Yamamoto [6,7], Ken Nakano [4] & Shoichi Kutsumizu [1]

Polymers that alter their properties and functions in response to carbon dioxide (CO$_2$) exposure offer significant potential for the development of smart technologies and innovative CO$_2$ utilization approaches. Nonetheless, effectively regulating the behavior of solid-state polymers using CO$_2$ remains a considerable challenge, highlighting the need for robust and reliable strategies to address this issue. This study presents elastomers that feature nanophase-separated morphologies composed of CO$_2$-vitrifiable polyethyleneimine and CO$_2$-permeable polydimethylsiloxane components. The elastomers (Young's modulus ($E$) of approximately 1 MPa) reversibly transform into hard plastics ($E > 2$ GPa) in the presence of CO$_2$. In addition to bulk stiffening, their surface adhesion and friction rapidly shift, and the material's fluorescence is significantly amplified. Here, we show that these multifunctional responses to CO$_2$ position the materials as innovative platforms for responsive mechanical systems and CO$_2$-activated optical devices, with potential applications in sensing, display, and data storage technologies.

Carbon dioxide (CO$_2$) is a non-toxic, biocompatible, and non-flammable carbon resource that is both inexpensive and widely available. However, because CO$_2$ is also a major greenhouse gas, its emissions require mitigation. The Intergovernmental Panel on Climate Change has acknowledged the severe impact of global warming, which is primarily driven by greenhouse gases, such as CO$_2$ released through human activities[1]. In response, researchers have been working to establish a carbon-recycling society by developing technologies for the separation[2,3], capture[4–7], storage[8,9], and utilization[10–22] of CO$_2$. Within the context of chemical CO$_2$ utilization, researchers have specifically focused on synthesizing polymers[10–14,17,18], methanol[15,16,20], concrete[19,22], and other materials[21] using CO$_2$ as a feedstock. Despite

ongoing efforts, the practical utilization of CO$_2$ remains markedly constrained, highlighting a critical need for the development of more diverse and efficient technologies to actively advance carbon-recycling and circular carbon-economy initiatives.

The use of CO$_2$ to control material functions has recently garnered attention in materials science, leading to the development of advanced materials. For instance, Miyasaka et al. recently synthesized metal–organic frameworks capable of switching their magnetic properties in response to CO$_2$ exposure[23–26]. Such CO$_2$-responsive materials are anticipated to contribute to the effective utilization of CO$_2$—a waste product—as a valuable resource. In the field of polymer chemistry, CO$_2$-responsive polymers have predominantly been

[1]Department of Chemistry and Biomolecular Science, Faculty of Engineering, Gifu University, Yanagido, Gifu, Japan. [2]Department of Engineering, Graduate School of Engineering, Gifu University, Yanagido, Gifu, Japan. [3]Department of Materials Science and Processing, Graduate School of Natural Science and Technology, Gifu University, Yanagido, Gifu, Japan. [4]Faculty of Environment and Information Sciences, Yokohama National University, Yokohama, Japan. [5]Graduate School of Medical Sciences Core Laboratory, Nagoya City University, Kawakami, Mizuho-cho, Mizuho-ku, Nagoya, Japan. [6]Department of Life Science and Applied Chemistry, Graduated School of Engineering, Nagoya Institute of Technology, Gokiso-cho, Showa-ku, Nagoya, Japan. [7]Present address: Department of Materials Chemistry, Faculty of Engineering, Shinshu University, Nagano, Japan. ✉e-mail: miwa.yohei.y6@f.gifu-u.ac.jp

developed as solutions, liquids, or nanoparticles[27–29]. Since the first report by George and Weiss[30], $CO_2$-responsive gels[30–38] and elastomers[39–44] have been developed as solid-state materials. However, the elastic modulus of these $CO_2$-responsive materials is limited to several megapascals. Consequently, developing solid-state materials whose mechanical properties can be broadly tuned on demand in response to $CO_2$ exposure represents a primary challenge in this field. For instance, a soft elastomer that rapidly transforms into a hard plastic upon exposure to $CO_2$ could serve not only as an advanced structural material but also as a tactile sensor for $CO_2$ detection or a smart coating that forms wear-resistant, low-friction, and easy-to-clean surfaces. Such innovative materials hold considerable potential for creating advanced smart products that leverage $CO_2$ to modulate their functions. However, the key ingredients enabling these materials to harden in response to $CO_2$ exposure remain unknown.

In this report, we present the design of a $CO_2$-curable polymer, wherein polyethyleneimine (PEI), serving as a $CO_2$-vitrifiable building block, is linked with telechelic epoxy polydimethylsiloxane (PDMS), exhibiting $CO_2$-permeable properties (Fig. 1a). Notably, PEI, a typical amine polymer, is unresponsive to $CO_2$ in the bulk state (Fig. 1b). However, it vitrifies upon $CO_2$ exposure when phase-separated at the nanoscale within a material. To control the size of this nanophase separation, herein, we used three telechelic epoxy PDMS variants (PDMS–H, PDMS–M, and PDMS–L) with different molecular weights. Among these, PDMS–H exhibits a number-average molecular weight ($M_n$) and polydispersity index (PDI) of 12,000 and 1.69, respectively (Supplementary Fig. 1). The corresponding values for PDMS–M and PDMS–L are 3800 and 1.75 and 1700 and 1.30, respectively. $CO_2$-curable polymers containing PDMS–H are designated as H($n$), where $n$ represents the weight percentage of PEI in the material. Similarly, those containing PDMS–M and PDMS–L are labeled as M($n$) and L($n$), respectively. In these $CO_2$-curable polymers, the continuous PDMS phase promotes $CO_2$ diffusion into the material, enabling rapid curing upon $CO_2$ exposure (Fig. 1a). Furthermore, this design allows on-demand tuning of mechanical properties—including Young's modulus, strength, toughness, and stretchability—depending on the composition of the $CO_2$-vitrifiable PEI component (Fig. 1c). For instance, the Young's modulus of the $CO_2$ curable polymers varies widely from approximately 2 MPa to approximately 2 GPa. Additionally, our findings indicate that these materials can function not only as mechanically responsive systems, but also as optical display and information recording frameworks utilizing $CO_2$ (Fig. 1d). They can also serve as high-performance $CO_2$ adsorbents, with capacities reaching approximately 5 mmol g$^{-1}$. The synthesized materials exhibit the dual functionality of $CO_2$ capture and utilization, while additionally conferring

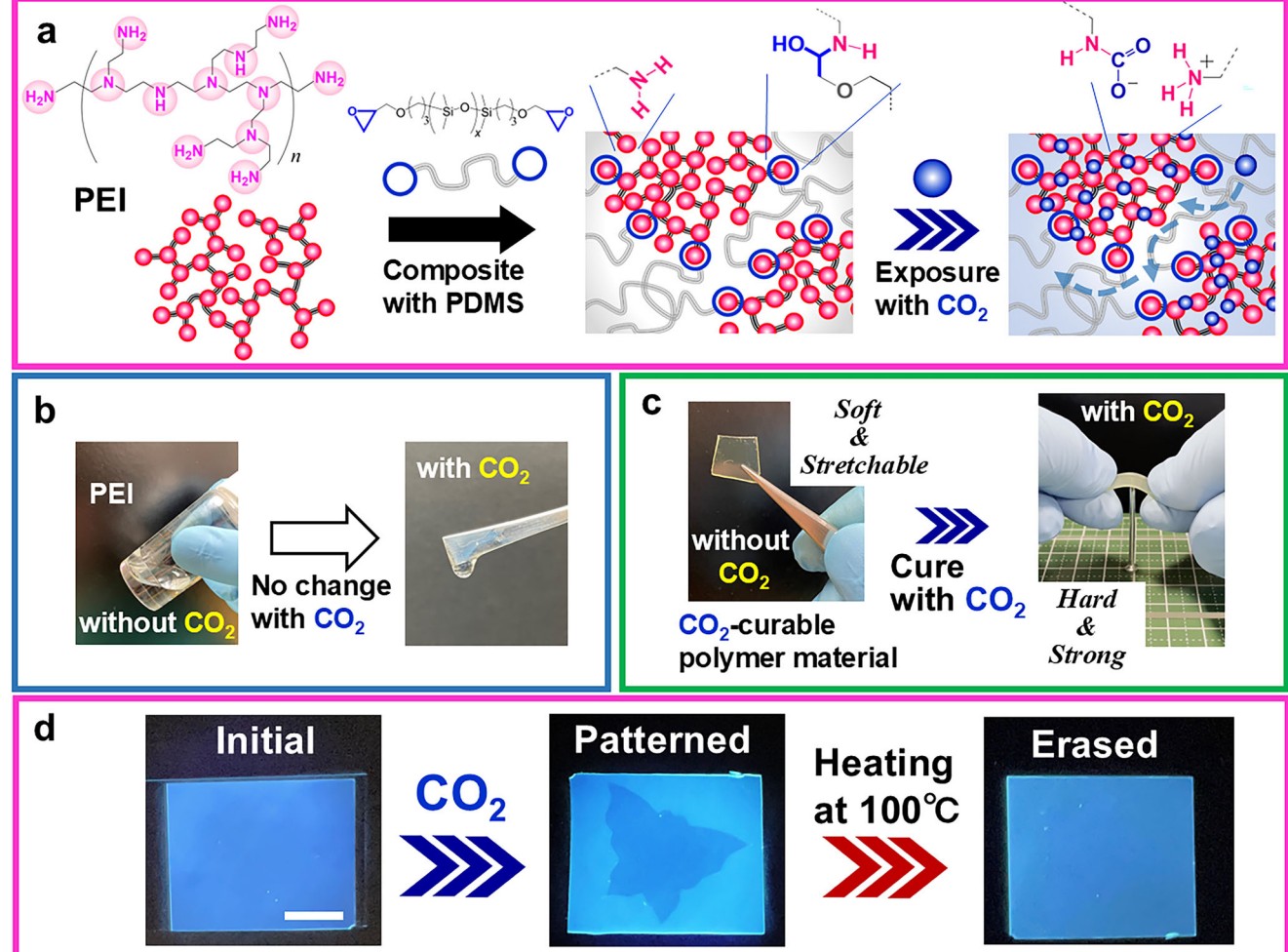

**Fig. 1 | Design concept of $CO_2$-curable polymers using PEI as the $CO_2$-vitrifiable building block. a** Schematic illustration of the preparation of $CO_2$-curable polymers and their interaction with $CO_2$ molecules. **b** Photographs of PEI, which remain unresponsive to $CO_2$ in the bulk state. **c** Photographs of $CO_2$-curable polymers with and without $CO_2$ exposure, illustrating their transformation from elastic to plastic behavior upon $CO_2$ exposure. **d** Photograph of an H(30) sheet under ultraviolet (UV) irradiation at 365 nm. Fluorescence intensity increases with $CO_2$ exposure. A butterfly-shaped pattern is recorded onto the sheet using an aluminum mask. The scale bar is 10 mm.

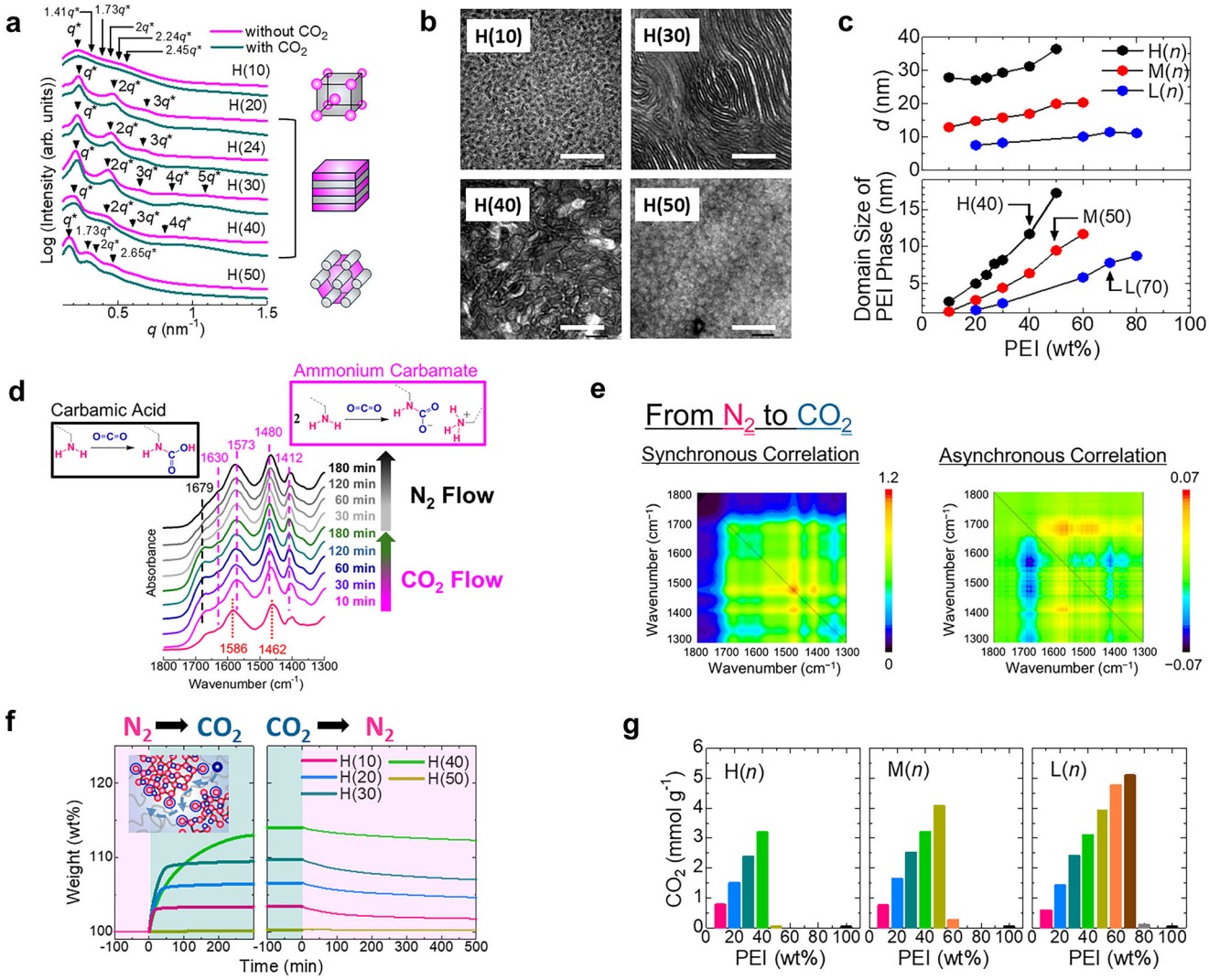

**Fig. 2 | Nanophase-separated morphology and $CO_2$ capture performance of the synthesized $CO_2$-curable polymers. a** SAXS profiles of H($n$) samples with and without $CO_2$ exposure. **b** TEM images of selected samples. PEI component was stained with $RuO_4$. The scale bar is 200 nm. **c** Average periodicity ($d$) and domain size of PEI phase as a function of PEI content. **d** Changes in the FTIR spectra of H(40) under $CO_2$ and $N_2$ flow. **e** Two-dimensional infrared correlation spectra recorded during the $N_2 \rightarrow CO_2$ process. **f** Weight changes of selected samples during gas switching between $N_2$ and $CO_2$ at 25 °C. **g** $CO_2$-capture capacity of the samples at 25 °C.

value to $CO_2$ through its capacity to modulate the mechanical and optical properties of the material.

## Results

### $CO_2$-capture performance

The $CO_2$-curable polymers synthesized in this study form transparent sheets regardless of the molecular weight of PDMS or the composition of PEI (Fig. 1c and Supplementary Fig. 2). Given the highly ordered nanophase separation between the PEI and PDMS phases, H($n$) samples exhibit multiple scattering peaks in their small-angle X-ray scattering (SAXS) profiles (Fig. 2a). In particular, the profiles of H(20), H(24), H(30), and H(40) contain integral multiples of $q^*$, indicative of a lamellar morphology. Here, $q^*$ denotes the scattering vector ($q$) of the primary scattering peak. Meanwhile, H(10) and H(50) exhibit spherical and cylindrical morphologies, respectively. The SAXS profiles of M($n$) and L($n$) samples are broad, which is attributed to the weak segregation of the low-molecular-weight PDMS used in these materials (Supplementary Fig. 3). The morphologies were further verified by transmission electron microscopy (TEM) (Fig. 2b and Supplementary Fig. 4). Overall, the SAXS analysis confirms that the morphology of the

materials remains unchanged following $CO_2$ exposure (Fig. 2a). The average domain size of the PEI phase, calculated from the average periodicity ($d$) and the volume fraction of PEI in each sample, decreases with the reduction in both the molecular weight of PDMS and PEI content (Fig. 2c). Here, the periodicity $d$ is determined using the relation $d = 2\pi \, q^{-1}$. In summary, the SAXS and TEM analyses reveal that the $CO_2$-curable polymers comprise a continuous PDMS phase that facilitates $CO_2$ diffusion alongside a nanoscale PEI phase, which collectively creates a large accessible interface between PEI and $CO_2$ molecules.

These polymers capture $CO_2$ through the formation of ammonium carbamate and carbamic acid, as illustrated in Fig. 2d[45]. Under $CO_2$ exposure, the bands at 1412, 1480, 1573, and 1630 $cm^{-1}$− attributed to the symmetric stretching of $COO^-$, symmetric deformation of $NH_3^+$, asymmetric stretching of $COO^-$, and asymmetric deformation of $NH_3^+$ in ammonium carbamates, respectively−in the Fourier-transform infrared (FTIR) spectrum of H(40) intensify (Fig. 2d). Simultaneously, the band at 1679 $cm^{-1}$, attributed to the C = O stretching of carbamic acid also becomes more prominent[45]. The change can be seen more clearly in the difference spectra (Supplementary Fig. 5). Conversely,

the band at 1586 cm$^{-1}$, assigned to the N–H bending of primary amine diminishes under $CO_2$ flow. The spectrum also induces a band at 1462 cm$^{-1}$, assigned to the C–H bending of methylene. Meanwhile, the synchronous correlation spectrum displays a positive cross peak between the bands at 1412 and 1480 cm$^{-1}$ during both switching processes, suggesting that they share a common origin (Fig. 2e). In contrast, the asynchronous correlation spectrum reveals that the bands at 1412, 1480, 1573, and 1630 cm$^{-1}$ exhibit negative cross peaks relative to the band at 1679 cm$^{-1}$ during the $N_2 \rightarrow CO_2$ process. However, during the reverse process ($CO_2 \rightarrow N_2$), these bands exhibit positive cross peaks relative to the same band (Supplementary Fig. 6). Collectively, these results indicate that ammonium carbamate generates before carbamic acid during the $N_2 \rightarrow CO_2$ process. In contrast, during the $CO_2 \rightarrow N_2$ process, ammonium carbamate exhibits greater stability and decomposes more slowly than carbamic acid.

The $CO_2$-capture performance of the materials was monitored by measuring weight changes during the $N_2 \rightarrow CO_2$ process. Although PEI captures only a small amount of $CO_2$ in its bulk state (Supplementary Fig. 7), the $CO_2$-curable polymers synthesized herein exhibit rapid and substantial $CO_2$ uptake owing to the combined effects of the continuous $CO_2$-permeable PDMS phase and nanosized PEI phase, which together generate a large accessible interface between PEI and $CO_2$ molecules (Fig. 2f and Supplementary Fig. 8). However, $CO_2$ release under $N_2$ flow at 25 °C proceeds slowly because the primary species formed during this reaction is stable ammonium carbamate (Fig. 2f). While this compound remains stable at room temperature, it decomposes upon heating (Supplementary Fig. 9). Furthermore, $CO_2$ uptake by the synthesized polymers increases proportionally with PEI content, reaching a maximum of approximately 5 mmol g$^{-1}$ for L(70) (Fig. 2g). Notably, this value is comparable to those of other high-performance amine-based $CO_2$ adsorbents[7]. For L(70), heating accelerates $CO_2$ uptake kinetics but reduces the overall uptake capacity (Supplementary Fig. 10). Moreover, the $CO_2$-capture performance of H(30) is maintained even after 10 cycles of $CO_2$ absorption and release by heating (Supplementary Fig. 11). Interestingly, $CO_2$ uptake by the synthesized polymers decreases sharply once the PEI content exceeds a threshold, corresponding to H(40), M(50), and L(70) for each series. Notably, this threshold increases as the molecular weight of PDMS decreases. Although the precise cause of this behavior remains unclear, we speculate that it is associated with the domain size of the PEI phase. This is because a thicker PEI phase hinders the accessibility of $CO_2$ molecules to the amine groups, as previously reported by Stafford et al.[46]. The measured domain sizes of PEI phase for H(40), M(50), and L(70) are 12 nm, 9.4 nm, and 7.8 nm, respectively (Fig. 2c). While these values are not identical owing to morphological heterogeneity, they are comparable.

## $CO_2$-curable polymers with tunable mechanical properties

The mechanical properties of $CO_2$-curable polymer materials improve substantially with $CO_2$ capture. For example, H(30) is soft and weak in the absence of $CO_2$ (Supplementary Movie 1). However, it hardens and strengthens upon $CO_2$ exposure (Supplementary Movie 2). This transformation is exemplified by the ability of a small $CO_2$-cured H(30) sample to lift a 2 kg weight. Further insights are provided by temperature-modulated differential scanning calorimetry (MDSC) analysis. Specifically, the MDSC trace of H(40) without $CO_2$ exposure reveals glass transition temperatures ($T_g$) of approximately −58 °C and −18 °C for PEI and PDMS components, respectively (Fig. 3a). The endothermic peak observed around −45 °C is attributed to the melting temperature of the PDMS component. Upon $CO_2$ exposure, the $T_g$ value of PEI component increases to approximately 72 °C, while that of PDMS component remains unchanged. The drastic increase in the $T_g$ value of PEI component is attributed to the formations of ammonium carbamates and carbamic acids that densely crosslink the PEI component (Fig. 1a and Fig. 2d). Notably, the $CO_2$-vitrified PEI component

consistently exhibits a $T_g$ value of approximately 72 °C regardless of the material's composition (Fig. 3a). In contrast, bulk PEI does not vitrify upon $CO_2$ exposure (Fig. 1b). In fact, the shear storage modulus of PEI increases only slightly under these conditions (Supplementary Fig. 12). PDMS is also unaffected by the presence of $CO_2$ (Supplementary Fig. 13). Due to the presence of the $CO_2$-vitrifiable PEI component, the H($n$), M($n$), and L($n$) samples exhibit curability upon exposure to $CO_2$. Notably, the tensile storage modulus ($E'$) of H(20), H(24), H(30), and H(40) increases substantially and rapidly during the $N_2 \rightarrow CO_2$ process (Fig. 3b). Moreover, the $CO_2$-curing of the materials can be observed even in ambient air (Supplementary Fig. 14). In contrast, the $E'$ values of the corresponding $CO_2$-cured samples exhibit only a slight decrease when held at 25 °C in an $N_2$ atmosphere. Following air exposure for one and two months, the Young's modulus ($E$) of $CO_2$-cured H(30) gradually declines from 470 MPa to 410 MPa and then to 250 MPa (Supplementary Fig. 15). This result indicates that the $CO_2$-curable polymers have some stability at room temperature. Nevertheless, the mechanical properties of the materials change more drastically under thermal stimuli. For instance, upon heating, the $CO_2$-cured samples soften rapidly owing to $CO_2$ release (Supplementary Fig. 16). As depicted in Fig. 3c, H(24) retains excellent mechanical cyclability during repeated cycles of $CO_2$ exposure and heating in $N_2$. Collectively, these findings indicate that $CO_2$-curable polymers are well suited for use as smart switching systems leveraging $CO_2$ as a trigger.

The $CO_2$-curable polymers synthesized herein transition from soft elastomers to hard, tough plastics in response to $CO_2$ exposure. Moreover, the modulus, stretchability, and toughness of $CO_2$-cured materials are broadly tunable through compositional adjustments (Fig. 3d and Supplementary Fig. 17). For instance, the $E$ of the $CO_2$-cured H($n$) samples increases from 2.4 MPa to 2.2 GPa as the $CO_2$-vitrifiable PEI content rises from 10 wt% to 40 wt% (Fig. 3e). In particular, the $E$ value of H(40) increases by more than 1500-fold following $CO_2$ exposure. Remarkably, the $CO_2$-cured H($n$) samples are approximately three orders of magnitude harder than previously reported $CO_2$-responsive polymers (Fig. 3f)[32,34–38,41,43]. Beyond modulus enhancement, both the stress at break ($\sigma_b$), and toughness of H($n$) samples also improve substantially upon $CO_2$ exposure (Fig. 3e). In particular, L(60) recorded high $\sigma_b$ value of 68 MPa (Supplementary Fig. 17). In this study, the toughness of the sample was characterized by the fracture energy, which was calculated by integrating the area under the stress-strain curve. In the $CO_2$-curable polymers, the PEI component vitrifies upon $CO_2$ exposure because of the formations of ammonium carbamates and carbamic acids that densely crosslink the PEI component. This is the mechanism for the drastic enhancement of mechanical properties of the materials. On the other hand, the $CO_2$-cured materials that contain a large amount of vitrified PEI lose their stretchability and become brittle. The toughness of the materials reaches its maximum at a PEI content of approximately 20–30% (Supplementary Fig. 17).

In addition to tunable mechanical performance, the synthesized materials exhibit sensitive and reversible changes in surface properties upon $CO_2$ exposure. For instance, an H(40) elastomer sheet displays some adhesion under ambient conditions; however, its surface hardens and loses its adhesive ability upon $CO_2$ exposure (Fig. 3g, Supplementary Fig. 18, and Supplementary Movie 3). This change is reversible, as heating to remove $CO_2$ quickly restores the adhesive property. A similar switching behavior is observed in surface friction (Fig. 3h). Specifically, the friction coefficient of an H(40) sheet varies markedly between approximately 0.9 and 0.1, in response to alternating $CO_2$ exposure and heating. Notably, compared to the bulk properties, such as modulus, of the synthesized materials (Fig. 3b), their surface characteristics—namely adhesion and friction—are considerably more sensitive to $CO_2$. Collectively, these results suggest that $CO_2$-curable polymers are well suited for use as smart coatings or skin layers capable of rapidly and reversibly creating wear-resistant, low-friction, and easy-to-clean surfaces upon $CO_2$ exposure.

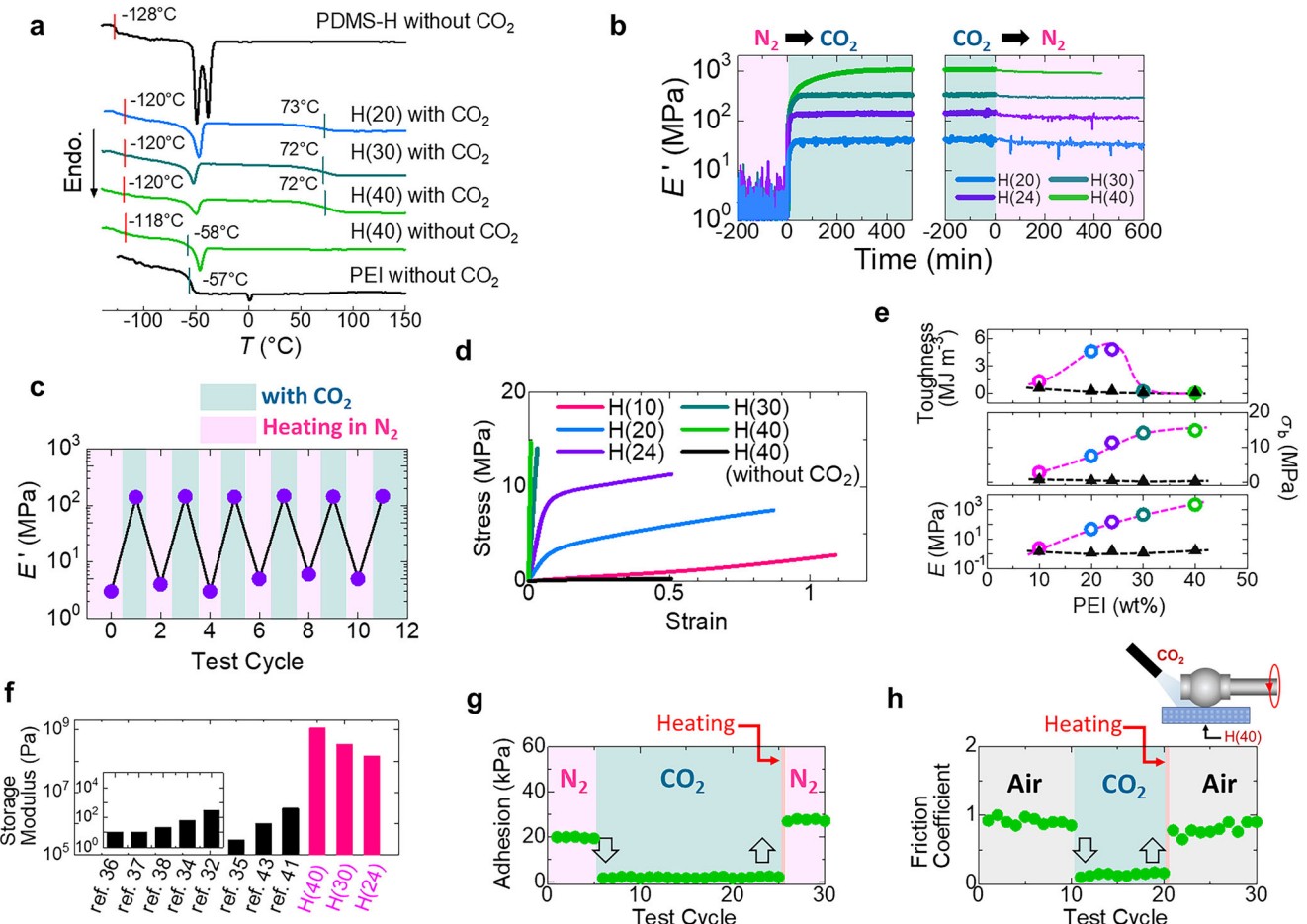

**Fig. 3 | Effect of CO₂ exposure on the mechanical properties of the synthesized CO₂-curable polymers. a** MDSC thermograms of selected samples. **b** Changes in the tensile storage moduli ($E'$) of selected H($n$) samples at 25 °C and 1 Hz during the N₂ → CO₂ and CO₂ → N₂ processes. **c** Cycling test results of H(24), involving CO₂ exposure followed by heating to 100 °C under N₂ flow. **d** Stress–strain curves obtained from uniaxial stretching tests of the selected samples. **e** Toughness,

stress at break ($\sigma_b$), and Young's modulus ($E$) of H($n$) samples as functions of PEI content. Colored and black plots correspond to samples with and without CO₂ exposure, respectively. **f** Comparison of the elastic moduli of H(24), H(30), and H(40) with those of conventional CO₂-responsive polymers. **g** Reversible change in the adhesion of H(40) at 25 °C upon CO₂ exposure followed by heating at 150 °C under N₂ flow. **h** Effect of CO₂ exposure on the surface friction of H(40).

## CO₂-switchable optical properties

The synthesized CO₂-curable polymers exhibit fluorescent luminescence under UV-irradiation at 365 nm in their initial state (Fig. 1d), owing to PEI crosslinking, as previously reported by Yang et al.[47]. Interestingly, this luminescence intensifies upon CO₂ exposure. As depicted in Fig. 4a, exposing H(40) to CO₂ substantially increases its UV absorption at 364 nm and fluorescence emission intensity at 460 nm (excited at 360 nm). This CO₂ exposure also results in slight increments in both quantum yield (Fig. 4b) and fluorescence lifetime ($\tau$) (Fig. 4c). This enhanced fluorescence is mainly attributed to the formation of ammonium carbamate[48]. Importantly, this property enables direct visualization of CO₂ diffusion into the material (Fig. 4d). Consistent with the results indicated by the gravimetric analysis (Fig. 2f), the rate of CO₂ diffusion in CO₂-curable polymers is visually observed to increase with increasing PDMS content (Fig. 4e). This is attributed to the CO₂ diffusion pathways facilitated by the highly CO₂-permeable PDMS component. The CO₂-enhanced fluorescence also facilitates patterning on the material's surface (Fig. 1d and Fig. 4a). Moreover, the pattern can be erased by releasing CO₂ through heating (Fig. 1d). Thus, this system demonstrates immense potential for diverse applications, including CO₂ detection, display technologies, and information recording systems.

## Discussion

In this study, we present innovative CO₂-curable polymers capable of dynamically adjusting their mechanical and optical properties in response to CO₂ exposure. A key component of this material design is a nanophase-separated morphology composed of CO₂-vitrifiable PEI and CO₂-permeable flexible PDMS phases, which enable rapid and reversible transitions in mechanical behavior—ranging from soft and stretchable to tough and hard. In addition to serving as advanced structural materials, these systems are also effective as smart skin materials with switchable adhesion and friction. Furthermore, they exhibit optical responsiveness due to enhanced fluorescence upon CO₂ exposure, making them capable of leveraging CO₂ for information display and recording. Collectively, these findings illustrate that CO₂-curable polymers hold promise not only as innovative structural platform but also as dynamic surface and optical systems, offering new directions for CO₂ utilization.

Importantly, the use of PEI as a CO₂-vitrifiable component is expected to broadly impart both mechanical and optical CO₂-responsiveness to a wide range of polymers. Furthermore, the combination of PEI with other polymers may enable precise control over these mechanical and optical properties, paving the way for the development of innovative functional materials. Ongoing research related to this topic is currently being conducted in our laboratory.

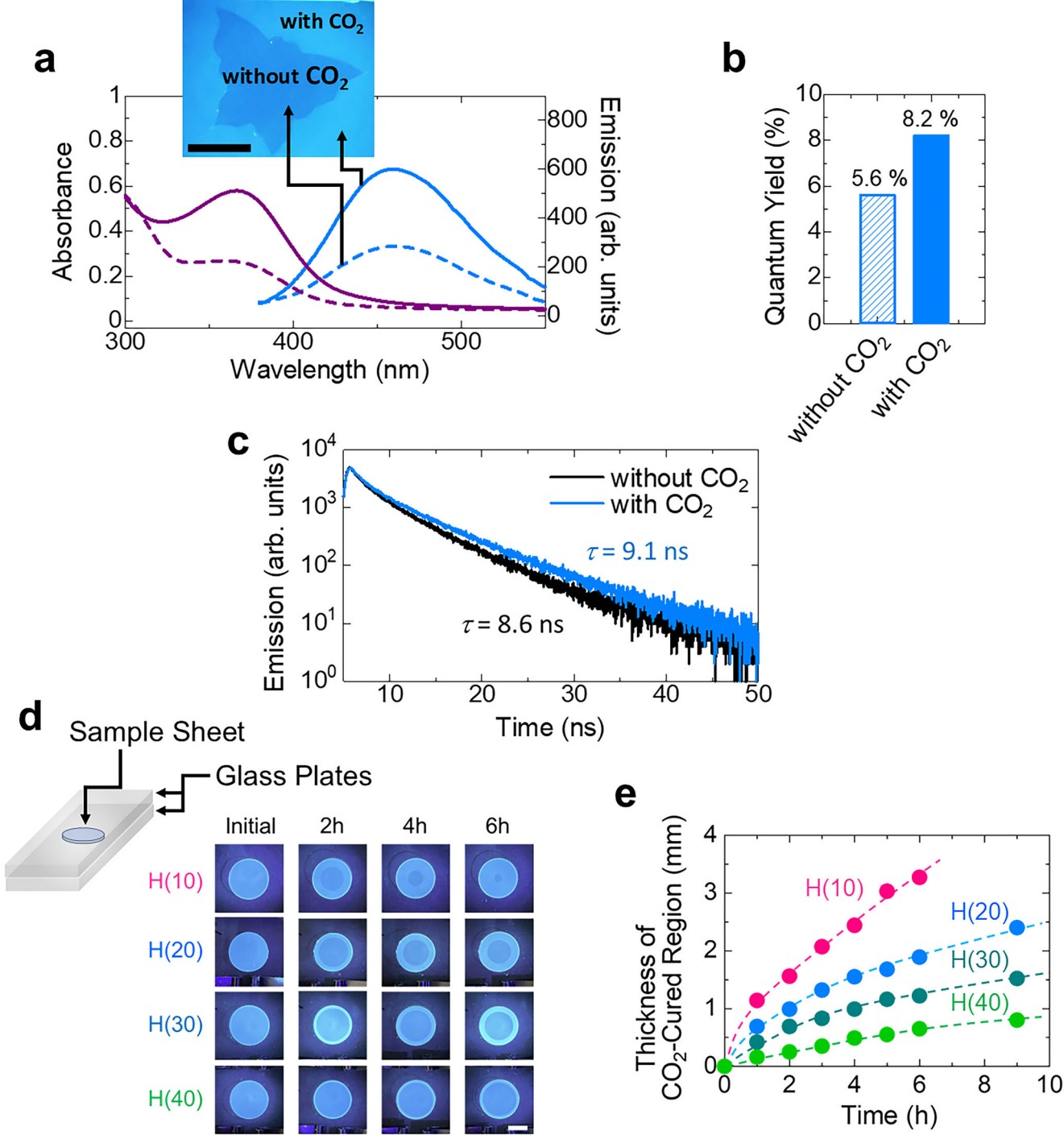

**Fig. 4 | CO₂-modulated fluorescence properties of H(40). a** UV absorption (purple) and fluorescence emission (blue) spectra of H(40) at 25 °C. Solid and dashed lines indicate spectra with and without CO₂ exposure, respectively. The excitation wavelength is 360 nm. The scale bar is 10 mm. **b** Impact of CO₂ exposure on the quantum yield (excited at 360 nm). **c** Fluorescence decay curves of H(40) with and without CO₂ exposure, measured under 360 nm excitation. The fluorescence lifetimes ($\tau$) of H(40) with and without CO₂ exposure are 8.6 and 9.1 ns, respectively. **d** Visualization of CO₂ diffusion into selected H($n$) sheets under UV irradiation. Here, circular sheets (8 mm in diameter) were sandwiched between glass plates and exposed to CO₂. The scale bar is 4 mm. **e** Thickness of the CO₂-cured region in the synthesized materials as a function of time.

## Methods

### Materials

Three kinds of telechelic epoxy polydimethylsiloxane (PDMS–H, PDMS–M, and PDMS–L) were obtained from Shin-Etsu Chemical Co., Ltd. and Fuzhou Topda New Material Co., Ltd (Supplementary Figs. 1, 19). Molecular sieves (3 A and 4 A, 1/16) and CDCl₃ (99.8%) was purchased from Nacalai Tesque, Inc. Chloroform (>99.0%)was obtained from Kanto Chemical Co., Inc. Branched polyethyleneimine (PEI) (number-average molecular weight ($M_n$) ≈ 10,000) was purchased from Junsei Chemical Co., Ltd. The ratio of primary, secondary, and tertiary amines of PEI determined by [13]C-NMR was 0.41, 0.32, and 0.27, respectively (Supplementary Fig. 20).

## Synthesis of H($n$), M($n$), and L($n$) samples

H($n$) samples, comprising $n$ wt% PEI linked with telechelic epoxy PDMS (PDMS−H, $M_n$ = 12,000; PDI = 1.69), were synthesized as follows: Weighed amounts of PEI and PDMS−H were dissolved in dehydrated chloroform to yield a solution with total concentration of 8 wt%. The solution was maintained under an argon atmosphere and stirred at 60 °C for 5 days. It was then transferred into a Teflon petri dish and slowly dried at 50 °C for 24 h to form an approximately 1-mm thick sample sheet. This sample sheet was further dried at 120 °C for 18 h under vacuum. The chemical composition of the samples was checked by FTIR (Supplementary Fig. 21). M($n$) and L($n$) samples were prepared similarly using PDMS−M ($M_n$ = 3800; PDI = 1.75) and PDMS−L ($M_n$ = 1,700; PDI = 1.30), respectively. The resulting sheets were transparent (Supplementary Fig. 2).

## Measurements

$^1$H-NMR and $^{13}$C-NMR measurements of samples dissolved in CDCl$_3$ containing tetramethylsilane were performed using a 400 MHz JEOL-ECS400 spectrometer.

An EXTREMA HPLC system (JASCO Co.), equipped with a polystyrene gel column (Shodex GPC LF-804), was employed for GPC measurements. THF was used as the eluent at 40 °C. PDMS standards (Scientific Polymer Products Inc.) were used for calibration.

An FT/IR-6600 spectrometer (JASCO Co.) was used for Fourier transform infrared (FTIR) measurements at a resolution of 4 cm$^{-1}$. The transmittance mode was used for samples deposited on a KBr plate. A 2D correlation program (JASCO) was used to generate the 2D infrared correlation spectra.

Thermogravimetric (TG-DSC) measurements were performed on an STA200 (HITACHI) with a gas flow of 100 mL min$^{-1}$.

Temperature-modulated differential scanning calorimetry (MDSC) was performed on an EXSTAR DSC6200 (Seiko Instruments Inc. [SII]). The samples were sealed in an aluminum pan and heated at a rate of 3 °C min$^{-1}$ from −135 °C to 180 °C. A modulation amplitude of 1.5 °C and a period of 60 s were used. A reversing heat flow was applied for analysis.

For sheet samples, the MCR702e instrument (Anton Paar) was employed for monitoring of modulus during gas switching. Tensile measurements were conducted at 1 Hz with a tensile strain of 0.1% at a constant temperature of 25 °C. The typical sample dimensions were 20 × 5.0 × 1.0 mm. Dry N$_2$ or CO$_2$ gas was flowed into the sample chamber at 3.0 L min$^{-1}$. For PEI, the MCR302 rheometer (Anton Paar) was employed. For the measurements, 1-mm-diameter parallel plates were used, and a shear strain of 0.03% was applied at 1 Hz. A Peltier cooling system was employed to maintain a constant temperature, and dry N$_2$ or CO$_2$ gas was flowed at 3.0 L min$^{-1}$ inside the Peltier hood.

Synchrotron SAXS measurements were performed using the BL8S3 beamline at the Aichi Synchrotron Radiation Center in Aichi, Japan. X-ray wavelength of 0.092 nm was used at a constant distance of approximately 2.2 m between the specimen and the detector. PILATUS 2 M was used as the detector. Silver behenate was used as the calibration standard. Before the measurement, some samples were fully exposed to CO$_2$ gas in an aluminum package. Each sample was then quickly installed into the beamline and measured in air.

The apparatus, a cryo unit (Leica FC7; Leica Microsystems) attached to an ultra-microtome (Leica UC7; Leica Microsystems), was used with liquid nitrogen. The sample pieces were frozen in the gas phase inside the cryo unit. A diamond knife (Syntek Co., Ltd.) was used to prepare ultrathin sections, with a thickness of approximately 100 nm. The sections were placed on copper grid with a carbon support film, for transmission electron microscope (TEM) observation. The sample grids were placed in a glass petri dish, and several drops of ruthenium tetroxide solution were added near the grids. The petri dish was covered, allowing the ruthenium to evaporate, and the specimens were stained. The stained ultrathin sections on the grid were observed using a TEM (JEM-1400Plus; JEOL Ltd.) at an acceleration voltage of 100 kV.

The Force Tester MCT-2150 (A&D Company, Ltd.) was used for the uniaxial tensile test. The test temperature was maintained at 26 °C ± 1 °C. JIS 7 dumbbell-shaped sample pieces were used for the tensile tests. Prior to the measurement, some specimens were fully exposed to CO$_2$ gas in an aluminum package for more than 2 days. The sample pieces were stretched at a strain rate of 0.09 s$^{-1}$. The toughness of the sample was characterized by the fracture energy, which was calculated by integrating the area under the stress-strain curve.

The Anton Paar MCR302 was used for the probe tack tests. A Peltier cooling system was employed to maintain a constant temperature, and dry N$_2$ or CO$_2$ gas was flowed at 3.0 L min$^{-1}$ inside the Peltier hood. The test was performed at 25 °C. A stainless-steel parallel plate with a diameter of 4 mm was used as the probe. An H(40) sheet was bonded onto a copper plate. The probe was brought into contact with the sample sheet with a load of 0.5 N for 5 s. Then, the probe was removed from the sample sheet at a speed of 10 μm s$^{-1}$. The measurements were repeated at 2 min intervals. After the measurements in CO$_2$, the sample was heated at 150 °C for 5 min under N$_2$ flow. After cooling, the measurement was performed again in N$_2$ atmosphere at 25 °C.

A custom-built ball-on-disk friction tester was used to evaluate the frictional properties of the sample. The system allowed CO$_2$ gas to be blown onto the sliding surface in air. Friction tests were conducted using a tribopair consisting of a square sample sheet and a stainless-steel ball (diameter: 19.05 mm). The sample sheet was mounted on a holder beneath the steel ball, which was rotated by a stepping motor, and the applied load was exerted by the steel ball onto the sample sheet. The sliding test conditions were as follows: an applied load of 1.0 N, a sliding speed of 1 mm s$^{-1}$, and a temperature of 25 °C. CO$_2$ gas was introduced at a flow rate of 1.0 L min$^{-1}$ onto the sliding surface. After the measurements in CO$_2$, the sample sheet was heated at 150 °C for 5 min in air using a rubber heater placed under the sample. After cooling, the measurement was performed again in air at 25 °C.

The UV absorption and fluorescent luminescence spectra were recorded using the UV-Vis spectrometer V-770 (JASCO Co.) and fluorescence spectrophotometer FP-8600 (JASCO Co.), respectively. H(40) was coated on a quartz glass plate to a thickness of ~0.1 mm and used for these measurements. The absolute PL quantum yield spectrometer C11347-01 (Hamamatsu Photonics) and fluorescence lifetime spectrometer Quantaurus-Qy C11367-01 (Hamamatsu Photonics) were used to obtain the quantum yield and the lifetime of fluorescent luminescence of H(40), respectively. The excitation wavelength was 360 nm.

## Data availability

All data are available in the main text and/or the Supplementary Information. All data are available from the corresponding author upon request.

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

## Acknowledgements

The authors thank Mr. K. Ohata at Graduate School of Natural Science and Technology, Gifu University for his experimental aids. Beam times at Aichi Synchrotron Radiation Center (20230511 and 202405049) are acknowledged herein. This research was financially supported by the JSPS KAKENHI (Grant Numbers 22H02141 (YM) and 25K01832 (YM)), Japan.

## Author contributions

Y.M. planned and directed the project. Y.M., K.O., T.H., K.Y., H.O., H.T., and K.H. conducted the experiments. Y.M., K.O., T.H., K.H., H.O., H.T., K.Y., K.N., and S.K. analyzed and discussed the data; Y.M., H.O., S.K., and K.O. wrote the paper. All authors have given approval to the final version of the manuscript.

## Competing interests

The authors declare no competing interests.
