## [Transparent Peer Review file · Nature Communications]

CO₂-triggered reversible transformation of soft elastomers into rigid and highly fluorescent plastics

Corresponding Author: Professor Yohei Miwa

Version 0:

Reviewer comments:

Reviewer #1

(Remarks to the Author)

This paper reports that elastomers prepared by crosslinking of polydimethylsiloxane (PDMS) with polyethyleneimine (PEI) exhibit significant changes in the mechanical properties under CO₂ atmosphere. PEI has the ability to absorb CO₂. It is also known that the composites of PDMS and amino polymers show increases in mechanical properties by exposing CO₂. However, the present paper reports extremely large changes of the elastic moduli from 1 MPa to 2 GPa. This is surprising. PDMS has excellent gas permeability, the reported composite polymers show fast response for CO₂. The structural characterization is sufficient. This paper seems worthy of being reported in Nature Communications.

The following points need to be revised.

- 1) Fig. 2d and Fig. 3g. These figures are too small to read the axis labels. How about moving some of these to the supporting information?
- 2) Fig. 3f. The reason why the modulus of PEI-PDMS composites is significantly larger than other reports should be discussed. This is the most important point of this paper. The authors seem to have enough data to provide an explanation.
- 3) Page 6, line 14. "offering considerable potential for advancing carbon recycling efforts." is an exaggeration.
- 4) Page 16, lines 14-15. "effectively utilize CO₂ – a waste product – as a valuable resource." is also an exaggeration. CCUS technology is significantly progressing, and there is no evidence that this PEI-PDMS composites are effective. To state "valuable resource", the cost calculations are required.
- 5) Page 10, lines 3-4. "This value is comparable to those of other high-performance CO₂ adsorbents" is an overestimation. There are many better adsorbents. The comparison should be made with amine-based materials.

Reviewer #2

(Remarks to the Author)

The combination of PDMS and PEI is commonly employed in CO₂ separation materials. In this work, the authors developed an elastomer composed of PDMS and PEI exhibiting a nanophase-separated morphology. The material demonstrates a reversible transformation between hard and soft plastic states under the control of CO₂ exposure and heating, along with tunable surface adhesion, frictional behavior, and fluorescence properties. The design for materials is delicate and the results are interesting. In view of these considerations, I recommend to publish this work in Nature Communications after addressing some concerns and problems.

1. The incorporation of PDMS is intended to enhance CO₂ permeability and promote interaction between CO₂ and PEI. However, no experimental data are provided to support the role of PDMS in influencing gas transport. It is recommended that the authors measure and report the CO₂ permeability of PDMS/PEI composites with varying PDMS ratios to better clarify the effect of PDMS content on gas permeation.
2. The nanophase-separated morphology of the PEI domains was estimated via SAXS analysis. However, to accurately characterize the "island-sea" morphology and the domain size, electron microscopy (e.g., TEM or SEM) may be necessary.

Such imaging could also provide visual evidence of the accessible interface between PEI and CO₂ molecules mentioned by the authors, and potentially explain the observed decline in CO₂ uptake with excess PEI content.

3. The CO₂-induced hardening of the material is indeed intriguing. Nevertheless, the potential practical application is difficult to envision, as the mechanical properties would deteriorate upon long-term air exposure due to CO₂ desorption. This raises concerns about the long-term stability of the material as a structural component.

4. The mechanical testing results confirm a substantial enhancement in strength after CO₂ uptake. While this may be attributed to the formation of ammonium carbamate and carbamic acid, it is important to note that such species may also lead to embrittlement or mechanical degradation under certain conditions. Additional experimental characterization or computational modeling would be helpful to elucidate the mechanism underlying the mechanical reinforcement induced by CO₂.

5. Fig. 4f is not present in the manuscript. In Line 150, "Fig. 4f" should be corrected to "Fig. 2f".

6. In Fig. 3c, both heating and N₂ atmosphere should be explicitly labeled. Otherwise, it may be misunderstood that rapid desorption is solely due to N₂ exposure.

7. In Line 110, the authors state that "the morphology of the materials remains unchanged following CO₂ exposure." However, the subsequent frictional performance tests indicate a noticeable change in the coefficient of friction. These two observations may appear contradictory and require clarification.

8. The authors claim that regardless of the molecular weight of PDMS or the composition of PEI, the resulting materials remain transparent. It would be helpful to include optical photographs of all samples to support this claim and aid reader understanding.

Reviewer #3

(Remarks to the Author)

1. Morphology Verification

Figure 2A presents SAXS profiles and fitted morphologies for the H-series. Is it possible to collect complementary imaging data—such as TEM or AFM—to independently verify the lamellar, spherical, and cylindrical domains inferred from the SAXS fits?

2. FTIR Spectral Intensities

The authors state that the bands at 1412, 1480, 1573, and 1630 cm⁻¹ "intensify" under CO₂ exposure. However, Figure 2c displays overlaid (stacked) rather than normalized spectra, making it difficult to judge relative intensity changes. Please provide normalized FTIR spectra or difference spectra to unambiguously demonstrate the reported band growth.

3. CO₂ Uptake Kinetics

Figure 2e shows that H(10), H(20), and H(30) reach different equilibrium weights but exhibit nearly identical initial uptake rates and equilibrate within 50 min. In contrast, H(40) displays slower kinetics and a longer equilibration time. What is the mechanistic rationale for this crossover at 40 wt% PEI? Do the M- and L-series show similar trends? Please include the full kinetic curves for M(n) and L(n) samples in the Supporting Information.

4. CO₂ Uptake Performance Focus

Figure 2f indicates that several M(n) and L(n) samples have higher CO₂ capacity than H(n). Please explain why subsequent experiments focus primarily on the H-series rather than on the materials with greater uptake performance.

5. PDMS Glass Transition Data

Lines 172–173 state that "upon CO₂ exposure, the T_g of PEI increases to ~72 °C, while that of PDMS remains unchanged." However, Figure 3A does not display the PDMS glass transition peak. Please add the bulk PDMS T_g data (before and after CO₂ exposure) or clarify if it was not measurable.

6. M(n) and L(n) Mechanical Data

Lines 177–179 note that "Due to the presence of the CO₂-vitrifiable PEI component, the H(n), M(n), and L(n) samples exhibit curability upon exposure to CO₂." However, no mechanical or rheological data or video for the M(n) or L(n) samples are provided in the main text or Supporting Information. Please provide representative curing curves (e.g., storage modulus vs. time under CO₂) for the M- and L-series.

7. Mechanical Properties Inconsistency

In Figure 3E, both H(30) and H(40) exhibit a sudden drop in toughness back to control levels, even as stress-at-break (σ_b) and Young's modulus (E) continue to increase. What is the origin of this inconsistency?

8. Discrepancy in Uptake and Curing Kinetics

Figure 2E shows nearly complete uptake within 50 min for most samples, yet Figure 4D reports CO₂-cured region growth over 9 hours, implying much slower diffusion. Were the samples in Figure 4D prepared differently (e.g., thicker films, different PEI content)? Please clarify the experimental conditions for each measurement and explain the apparent difference in rates.

9. Benchmarking Mechanical Modulus

Although the Introduction cites prior work on CO₂-responsive gels and elastomers (refs 39–44), it would strengthen the narrative to directly compare the maximum moduli achieved previously with the >2 GPa reported here.

10. CO₂ Capture under Cyclic Conditions

While the CO₂-responsiveness is interesting, it is not clear whether the material's CO₂ capture capacity is retained under

cyclic use (i.e., after repeated mechanical/optical switching).

11. Environmental Interference

Does ambient moisture or other gases (e.g., O₂) interfere with the observed switching or CO₂ uptake?

12. Editorial Corrections

- “primary driven by greenhouse gases” → “primarily driven”
- “to control the sized of this nanophase separation” → “size”

Figure 1 can be improved. It is crowded and poorly organised i.e. placement of a, b, c, d. Different fonts, text sizes and colours.

Figure 2 labels are unreadable.

Figure 4 caption replace "sloid" with solid. Fluorescence decays should be presented rather than a bar chart.

Reviewer #4

(Remarks to the Author)

Version 1:

Reviewer comments:

Reviewer #1

(Remarks to the Author)

[Editor's note: This reviewer left remarks to editor, and does not have any further technical concerns]

Reviewer #2

(Remarks to the Author)

The authors added some extra experimnts and characterizations to clarify the concerns, and they have well addressed almost all of the questions. In view of these, I recommend to publish this work in its present form.

Reviewer #3

(Remarks to the Author)

I am satisfied that all points raised by the reviewers have been sufficiently answered in the revised manuscript. My recommendation is to accept in the current format.

Reviewer #4

(Remarks to the Author)

According to the reviewer's comments, we revised our manuscript, and our answers to the reviewer's questions are as follows:

Comments from Reviewer #1:

Comments :

This paper reports that elastomers prepared by crosslinking of polydimethylsiloxane (PDMS) with polyethyleneimine (PEI) exhibit significant changes in the mechanical properties under CO₂ atmosphere. PEI has the ability to absorb CO₂. It is also known that the composites of PDMS and amino polymers show increases in mechanical properties by exposing CO₂. However, the present paper reports extremely large changes of the elastic moduli from 1 MPa to 2 GPa. This is surprising. PDMS has excellent gas permeability, the reported composite polymers show fast response for CO₂. The structural characterization is sufficient. This paper seems worthy of being reported in Nature Communications.

The following points need to be revised.

Thank you for the reviewer's constructive comments and kind suggestions. These have been helpful in improving the quality of our work. In accordance with the reviewer's feedback, we have revised the manuscript. Point-by-point responses to the comments are provided below.

1) Fig. 2d and Fig. 3g. These figures are too small to read the axis labels. How about moving some of these to the supporting information?

Response: Fig. 2d. and Fig. 3g have been partially moved to Supplementary Fig. 6 and Supplementary Fig. 18, respectively.

2) Fig. 3f. The reason why the modulus of PEI-PDMS composites is significantly larger than other reports should be discussed. This is the most important point of this paper. The authors seem to have enough data to provide an explanation.

Response: Thank you for your insightful suggestion. In the PEI-PDMS composites, the PEI component vitrifies upon CO₂ exposure due to the formation of ammonium carbamates and carbamic acids, which densely crosslink the PEI chains. This is the mechanism responsible for the hardening of the material. In the revised manuscript, we have added the following sentences: (Page 11, line 181) **The drastic increase in the T_g value of PEI component is attributed to the formations of ammonium carbamates and carbamic acids that densely crosslink the PEI**

component (Fig. 1a and Fig. 2d); (Page 7, line 219) In the CO₂-curable polymers, the PEI component vitrifies upon CO₂ exposure because of the formations of ammonium carbamates and carbamic acids that densely crosslink the PEI component. This is the mechanism for the drastic enhancement of mechanical properties of the materials.

3) Page 6, line 14. “offering considerable potential for advancing carbon recycling efforts.” is an exaggeration.

Response: Thank you for the reviewer’s comments. We deleted the phrase and revised the sentence as follows: (Page 6, line 96) “The synthesized materials exhibit the dual functionality of CO₂ capture and utilization, while additionally conferring novel value to CO₂ through its capacity to modulate the mechanical and optical properties of the material.”

4) Page 16, lines 14-15. “effectively utilize CO₂ – a waste product – as a valuable resource.” is also an exaggeration. CCUS technology is significantly progressing, and there is no evidence that this PEI-PDMS composites are effective. To state “valuable resource”, the cost calculations are required.

Response: Thank you for the reviewer’s comments. We deleted the phrase and revised the sentence as follows: (Page 18, line 279) “Furthermore, the combination of PEI with other polymers may enable precise control over these mechanical and optical properties, paving the way for the development of innovative functional materials.”

5) Page 10, lines 3-4. “This value is comparable to those of other high-performance CO₂ adsorbents” is an overestimation. There are many better adsorbents. The comparison should be made with amine-based materials.

Response: Thank you for the reviewer’s comments. We revised the sentence as follows: (Page 10, line 154) “Notably, this value is comparable to those of other high-performance amine-based CO₂ adsorbents.”

Comments from Reviewer #2:

The combination of PDMS and PEI is commonly employed in CO₂ separation materials. In this work, the authors developed an elastomer composed of PDMS and PEI exhibiting a nanophase-separated morphology. The material demonstrates a reversible transformation between hard and soft plastic states under the control of CO₂ exposure and heating, along with tunable surface adhesion, frictional behavior, and fluorescence properties. The design for materials is delicate and the results are interesting. In view of these considerations, I recommend to publish this work in Nature Communications after addressing some concerns and problems.

We appreciate the reviewer’s positive feedback and thoughtful suggestions, which have been valuable in enhancing the quality of our work. In response to the reviewer’s comments, we have revised the manuscript accordingly. Point-by-point responses to the comments are presented below:

1) The incorporation of PDMS is intended to enhance CO₂ permeability and promote interaction between CO₂ and PEI. However, no experimental data are provided to support the role of PDMS in influencing gas transport. It is recommended that the authors measure and report the CO₂ permeability of PDMS/PEI composites with varying PDMS ratios to better clarify the effect of PDMS content on gas permeation.

Response: Thank you for the indication. We consider that the data in Fig. 4d and Fig. 4e in the revised manuscript visually indicate enhanced CO₂ permeability in the CO₂-curable polymer materials by the PDMS component. On page 16, line 251 of the revised manuscript, we added the following sentence: “Consistent with the results indicated by the gravimetric analysis (Fig. 2f), the rate of CO₂ diffusion in CO₂-curable polymers is visually observed to increase with increasing PDMS content (Fig. 4e). This is attributed to the CO₂ diffusion pathways facilitated by the highly CO₂-permeable PDMS component.”

2) The nanophase-separated morphology of the PEI domains was estimated via SAXS analysis. However, to accurately characterize the “island-sea” morphology and the domain size, electron microscopy (e.g., TEM or SEM) may be necessary. Such imaging could also provide visual evidence of the accessible interface between PEI and CO₂ molecules mentioned by the authors, and potentially explain the observed decline in CO₂ uptake with excess PEI content.

Response: Thank you for the suggestion. We observed the morphologies of H(10), H(30), H(40), H(50), and M(50) using TEM. The TEM images are shown in Fig. 2b and Supplementary Fig. 4. The observed morphologies of the H(*n*) samples are consistent with those determined by SAXS, whereas the morphology of M(50) is not clearly visible in the TEM images. On page 7, line 111 of the revised manuscript, we added the following sentence: “The morphologies were further verified by transmission electron microscopy (TEM) (Fig. 2b and Supplementary Fig. 4).”

For H(30), where a highly ordered morphology is observed by TEM, we confirmed that the domain sizes of the PEI phase determined by SAXS and TEM are in good agreement. On the other hand, the domain size of the PEI phase is difficult to determine from the TEM images of samples without highly ordered morphology, such as M(50). Therefore, we believe that the average domain size of the PEI phase should be determined from SAXS.

3) The CO₂-induced hardening of the material is indeed intriguing. Nevertheless, the potential practical application is difficult to envision, as the mechanical properties would deteriorate upon

long-term air exposure due to CO₂ desorption. This raises concerns about the long-term stability of the material as a structural component.

Response: Thank you for the reviewer's comments. The data on the long-term stability of the material are shown in Supplementary Fig. 16 of the revised Supplementary Information. We examined the mechanical properties of CO₂-cured H(30) stored for one and two months at room temperature in air. As the reviewer pointed out, the Young's modulus of CO₂-cured H(30) decreased during storage. However, the decrease was gradual. CO₂-cured H(30) retained 87% and 53% of its initial Young's modulus after one and two months, respectively. On page 12, line 197 of the revised manuscript, we have added the following sentence: "This result indicates that the CO₂-curable polymers have some stability at room temperature."

4) The mechanical testing results confirm a substantial enhancement in strength after CO₂ uptake. While this may be attributed to the formation of ammonium carbamate and carbamic acid, it is important to note that such species may also lead to embrittlement or mechanical degradation under certain conditions. Additional experimental characterization or computational modeling would be helpful to elucidate the mechanism underlying the mechanical reinforcement induced by CO₂.

Response: Thank you for the reviewer's insightful suggestion. As the reviewer pointed out, the enhancement of mechanical properties of the materials is attributed to the formation of ammonium carbamates and carbamic acids. On page 14, line 219 of the revised manuscript, we stated as follows: "In the CO₂-curable polymers, the PEI component vitrifies upon CO₂ exposure because of the formations of ammonium carbamates and carbamic acids that densely crosslink the PEI component. This is the mechanism for the drastic enhancement of mechanical properties of the materials." Moreover, CO₂-cured materials containing a high amount of vitrified-PEI become poorly stretchable and brittle. To demonstrate this property, we plotted the toughness of the CO₂-cured materials in Supplementary Fig. 17c. The toughness of the materials is at a maximum at about 20% to 30% PEI. On page 14, line 222 of the revised manuscript, we stated as follows: "On the other hand, the CO₂-cured materials that contain a large amount of vitrified PEI lose their stretchability and become brittle. The toughness of the materials reaches its maximum at a PEI content of approximately 20–30% (Supplementary Fig. 17)."

5) Fig. 4f is not present in the manuscript. In Line 150, "Fig. 4f" should be corrected to "Fig. 2f".

Response: Thank you for the indication. We corrected the figure number.

6) In Fig. 3c, both heating and N₂ atmosphere should be explicitly labeled. Otherwise, it may be misunderstood that rapid desorption is solely due to N₂ exposure.

Response: We changed the label in Fig. 3c as follows:

7) In Line 110, the authors state that “the morphology of the materials remains unchanged following CO₂ exposure.” However, the subsequent frictional performance tests indicate a noticeable change in the coefficient of friction. These two observations may appear contradictory and require clarification.

Response: Thank you for the reviewer’s question. We believe that the surface friction of H(40) is switchable without any morphological change in the material, because the driving force behind the change in friction coefficients is the CO₂-induced vitrification of the PEI component.

8) The authors claim that regardless of the molecular weight of PDMS or the composition of PEI, the resulting materials remain transparent. It would be helpful to include optical photographs of all samples to support this claim and aid reader understanding.

Response: Thank you for the reviewer’s indication. We exhibit photographs of all samples in Supplementary Fig. 2.

Comments from Reviewer #3:

1) Morphology Verification

Figure 2A presents SAXS profiles and fitted morphologies for the H-series. Is it possible to collect complementary imaging data—such as TEM or AFM—to independently verify the lamellar, spherical, and cylindrical domains inferred from the SAXS fits?

Response: We observed the morphologies of selected H(*n*) samples using TEM. The TEM images are shown in Fig. 2b of the revised manuscript. The observed morphologies of H(10), H(30), H(40), and H(50) are spherical, lamellar, lamellar, and cylindrical, respectively. These morphologies are consistent with those identified by SAXS.

2) FTIR Spectral Intensities

The authors state that the bands at 1412, 1480, 1573, and 1630 cm^{-1} “intensify” under CO_2 exposure. However, Figure 2c displays overlaid (stacked) rather than normalized spectra, making it difficult to judge relative intensity changes. Please provide normalized FTIR spectra or difference spectra to unambiguously demonstrate the reported band growth.

Response: Thank you for the reviewer’s advice. In the revised manuscript, we added difference spectra in Supplementary Fig. 5. The difference spectra clearly exhibit the increase in the

intensity of the bands at 1412, 1480, 1573, 1630, and 1679 cm^{-1} . Also, on page 8, line 128 of the revised manuscript, we added a sentence as follows: **The change can be seen more clearly in the difference spectra (Supplementary Fig. 5).**

3) CO_2 Uptake Kinetics

Figure 2e shows that H(10), H(20), and H(30) reach different equilibrium weights but exhibit nearly identical initial uptake rates and equilibrate within 50 min. In contrast, H(40) displays slower kinetics and a longer equilibration time. What is the mechanistic rationale for this crossover at 40 wt% PEI? Do the M- and L-series show similar trends? Please include the full kinetic curves for M(n) and L(n) samples in the Supporting Information.

Response: We added the CO_2 uptake curves for all M(n) and L(n) samples in Supplementary Fig. 8. As shown in Fig. 2g, these samples exhibit a threshold behavior in their CO_2 uptake performance. The CO_2 uptake rate of all sample series slows down when the PEI content approaches this threshold. Therefore, we consider that an increase in the domain size of the PEI phase leads to slower CO_2 uptake in the PEI phase.

4) CO_2 Uptake Performance Focus

Figure 2f indicates that several M(n) and L(n) samples have higher CO_2 capacity than H(n). Please explain why subsequent experiments focus primarily on the H-series rather than on the materials with greater uptake performance.

Response: Thank you for the suggestion. In this study, our research primarily focuses on switching the mechanical properties of the materials in response to CO_2 . As shown in

Supplementary Fig. 17c, CO₂-cured H(n) samples exhibit not only a high modulus but also good stretchability compared to the M(n) and L(n) samples. Therefore, H(n) samples were primarily emphasized in this work.

5) PDMS Glass Transition Data

Lines 172–173 state that “upon CO₂ exposure, the T_g of PEI increases to ~72 °C, while that of PDMS remains unchanged.” However, Figure 3A does not display the PDMS glass transition peak. Please add the bulk PDMS T_g data (before and after CO₂ exposure) or clarify if it was not measurable.

Response: We measured MDSC of PDMS with and without CO₂. The T_g of PDMS is –128 °C in both cases. We added MDSC data for PDMS in Fig. 3a and Supplementary Fig. 13 in the revised manuscript. Also, on page 12, line 188 of the revised manuscript, we added the following sentence: **PDMS is also unaffected by the presence of CO₂ (Supplementary Fig. 13).**

6) M(n) and L(n) Mechanical Data

Lines 177–179 note that “Due to the presence of the CO₂-vitrifiable PEI component, the H(n), M(n), and L(n) samples exhibit curability upon exposure to CO₂.” However, no mechanical or rheological data or video for the M(n) or L(n) samples are provided in the main text or

Supporting Information. Please provide representative curing curves (e.g., storage modulus vs. time under CO₂) for the M- and L-series.

Response: Tensile test data for M(*n*) and L(*n*) samples cured with CO₂ are shown in Supplementary Fig. 17. Here, we provided tensile data instead of DMA data because the tensile tests reveal not only the modulus, but also the stretchability and strength of the samples.

7) Mechanical Properties Inconsistency

In Figure 3E, both H(30) and H(40) exhibit a sudden drop in toughness back to control levels, even as stress-at-break (σ_b) and Young's modulus (*E*) continue to increase. What is the origin of this inconsistency?

Response: As described in the Methods section, the toughness of the sample was characterized by the fracture energy, which was calculated by integrating the area under the stress-strain curve. H(30) and H(40) exhibit high stress-at-break and Young's modulus values. On the other hand, the strain-at-break of these samples is very low. Therefore, the toughness values of these samples are also low. On page 14, line 217 of the revised manuscript, we stated the following: "In this study, the toughness of the sample was characterized by the fracture energy, which was calculated by integrating the area under the stress-strain curve."

8) Discrepancy in Uptake and Curing Kinetics

Figure 2E shows nearly complete uptake within 50 min for most samples, yet Figure 4D reports CO₂-cured region growth over 9 hours, implying much slower diffusion. Were the samples in Figure 4D prepared differently (e.g., thicker films, different PEI content)? Please clarify the experimental conditions for each measurement and explain the apparent difference in rates.

Response: We used the same sample sheets for the measurements shown in Fig. 2f and Fig. 4e. Moreover, the CO₂ diffusion rates in the sample sheets, estimated from Fig. 2f and Fig. 4e, are nearly identical.

The sample sheets are approximately 1 mm thick, as described in the Methods section. In the TG-DSC measurement shown in Fig. 2f, CO₂ penetrates from both sides of the sheet. For example, in the case of H(20), CO₂ takes approximately 50 minutes to penetrate to a depth of 0.5 mm from the surface.

In contrast, in the setup used for Fig. 4e, the planar surfaces of the sheet are covered with glass plates, as illustrated in Fig. 4d. In this setup, the maximum depth from the side surface is 4 mm, since the diameter of the disc-shaped sample sheet is 8 mm. As shown in Fig. 4e, CO₂ takes approximately 50 minutes to penetrate 0.5 mm into H(20).

Therefore, we conclude that the CO₂ penetration rates into the sample sheets, as estimated from Fig. 2f and Fig. 4e, are essentially the same.

9) Benchmarking Mechanical Modulus

Although the Introduction cites prior work on CO₂-responsive gels and elastomers (refs 39–44), it would strengthen the narrative to directly compare the maximum moduli achieved previously with the >2 GPa reported here.

Response: We agree with the reviewer's suggestion. Therefore, we compared the elastic moduli of our samples with those of previously reported CO₂-responsive polymers in Fig. 3f.

10) CO₂ Capture under Cyclic Conditions

While the CO₂-responsiveness is interesting, it is not clear whether the material's CO₂ capture capacity is retained under cyclic use (i.e., after repeated mechanical/optical switching).

Response: In Supplementary Fig. 11 of the revised manuscript, we added new data on the CO₂-capture capability of H(30) after cyclic use. As shown in Supplementary Fig. 11, the CO₂-capture capability of H(30) remains unchanged even after 10 cycles of use. On page 10, line 157 of the revised manuscript, we stated the following: "Moreover, the CO₂-capture performance of H(30) is maintained even after 10 cycles of CO₂ absorption and release by heating (Supplementary Fig. 11)."

11) Environmental Interference

Does ambient moisture or other gases (e.g., O₂) interfere with the observed switching or CO₂ uptake?

Response: To estimate the effect of ambient moisture and oxygen on CO₂ curing, we monitored the storage modulus (E') of H(24) at 25 °C during the ambient air → CO₂ transition. The relative humidity was 50%. The result is shown in Supplementary Fig. 14 of the revised manuscript. CO₂ curing behavior of H(24) was observed even in air, and the effect of ambient moisture and oxygen on the CO₂ curing was found to be extremely small. On page 12, line 192 of the revised manuscript, we added the following comment:

Moreover, the CO₂-curing of the materials can be observed even in ambient air (Supplementary Fig. 14).

12) Editorial Corrections

- “primary driven by greenhouse gases” → “primarily driven”
- “to control the sized of this nanophase separation” → “size”

Response: Thank you. We corrected them.

Figure 1 can be improved. It is crowded and poorly organised i.e. placement of a, b, c, d. Different fonts, text sizes and colours.

Response: We corrected them. We changed the placement of panels, fonts, text sizes.

Figure 2 labels are unreadable.

Response: We corrected Fig. 2.

Figure 4 caption replace "sloid" with solid. Fluorescence decays should be presented rather than a bar chart.

Response: We corrected them.

Comments from Reviewer 4:

Thank you for reviewing our manuscript.

We hope that the manuscript is acceptable now.

Sincerely,

Yohei Miwa

Professor

Department of Chemistry and Biomolecular Science,

Faculty of Engineering, Gifu University

Yanagido, Gifu 501-1193, Japan.

Telephone: +81-58-293-2565

FAX: +81-58-293-2565

E-mail: miwa.yohei.y6@f.gifu-u.ac.jp